# Detection of Hepatic Metastasis from Early Delayed Images of Modified Dual-Time-Point F-18 FDG PET/CT Images in a Patient with Breast Cancer

**DOI:** 10.3390/diagnostics14151653

**Published:** 2024-07-31

**Authors:** Ji Young Lee, Hee-Sung Song

**Affiliations:** Department of Nuclear Medicine, Jeju National University Hospital, Jeju National University School of Medicine, Jeju-si 63241, Republic of Korea; easy02000@naver.com

**Keywords:** fluorodeoxyglucose, positron emission tomography/computed tomography, breast cancer, modified dual-phase, hepatic metastasis

## Abstract

We present a rare case of focal F-18-2-fluoro-2-deoxyglucose (FDG) uptake in the liver observed during a modified dual-time-point F-18 FDG positron emission tomography (PET)/computed tomography (CT), so-called early delayed scanning, in a 53-year-old woman diagnosed with breast cancer. This metastatic lesion was revealed in 80 min delayed images after FDG injection, but not in the usual one-hour images after injection. Modified dual-time-point F-18 FDG PET/CT is convenient because compared to the 2 h delayed images of dual-time-point PET/CT, it has a shorter scanning time and avoids additional radiation exposure.

A 53-year-old woman with right-breast cancer underwent F-18-2-fluoro-2-deoxyglucose (FDG) positron emission tomography (PET)/computed tomography (CT) as part of a metastatic workup. She was diagnosed with invasive ductal carcinoma in March 2020 and underwent breast-conserving surgery with axillary dissection. At the time of diagnosis, the pathological stage was IIA (T1N1M0); postoperative chemoradiotherapy was continued until November 2020.

In July 2023, a routine abdominopelvic CT revealed a new ill-defined high-attenuation lesion in segment 7 of the liver. Laboratory test results, including tumor and inflammatory marker levels, were within normal ranges. Consequently, the patient underwent modified dual-time-point (DTP) F-18 FDG PET/CT for further evaluation of the hepatic lesions. A focal hypermetabolic lesion was identified on routine 60 min images post-F-18 FDG injection in segment 7 of the liver, similar to the lesions seen on CT images (Figure 1a). Our department performed a modified, delayed scan 80 min after F-18 FDG injection, and another small increase in focal FDG uptake at the subcapsular portion of segment 6 of the liver (Figure 1d), in addition to the existing segment 7 lesion (Figure 1c), was detected. Although this lesion had a mild degree of FDG uptake, it was observed on several slices, unlike the surrounding heterogeneous hepatic uptake (Figure 2). This additional lesion was not observed on the early PET/CT images (Figure 1b); however, it was suspected to be metastatic. No other abnormal FDG uptakes suggesting metastasis or active inflammation were observed throughout the body.

Subsequent hepatic magnetic resonance imaging (MRI) was performed to assess the hypermetabolic lesions in the liver. On liver MRI, an arterial-enhancing lesion of approximately 2.3 cm with delayed washout in segment 7 of the liver (Figure 3a,b) and a faint arterial-enhancing lesion of approximately 0.8 cm in the subcapsular portion in segment 6 of the liver (Figure 3c,d) were observed; these were probably a hepatocellular carcinoma or metastases. Since the lesion in segment 6 was too small to be observed on liver ultrasound, an ultrasound-guided liver biopsy was performed only on the lesion of segment 8. The hepatic biopsy confirmed a metastatic carcinoma for which the patient received chemotherapy.

Many reports have indicated that DTP F-18 FDG PET may help distinguish benign from malignant lesions and evaluate uncertain metastatic lesions, particularly hepatic metastases; these are based on previous studies showing that FDG uptake increases in malignant tumors for several hours after injection, while uptake in benign lesions decreases or remains stable [1,2,3,4]. Modified DTP F-18 FDG PET/CT images performed 80 min after F-18 FDG injection, so-called early delayed images, have recently been reported in several articles [5,6,7]. This technique is advantageous because compared to the conventional DTP F-18 FDG PET/CT, wherein there is delayed acquisition at 120–180 min after FDG injection, it has a shorter scanning time and avoids additional radiation exposure. This is the first case report in which early delayed DTP F-18 FDG PET/CT was used to detect an additional metastatic lesion, in which only FDG uptake was observed on early delayed images. This conveniently modified DTP FDG PET/CT can help detect metastasis and recurrence in patients with cancer, especially in lesions with physiological FDG uptake such as lesions in the liver.

## Figures and Tables

**Figure 1 diagnostics-14-01653-f001:**
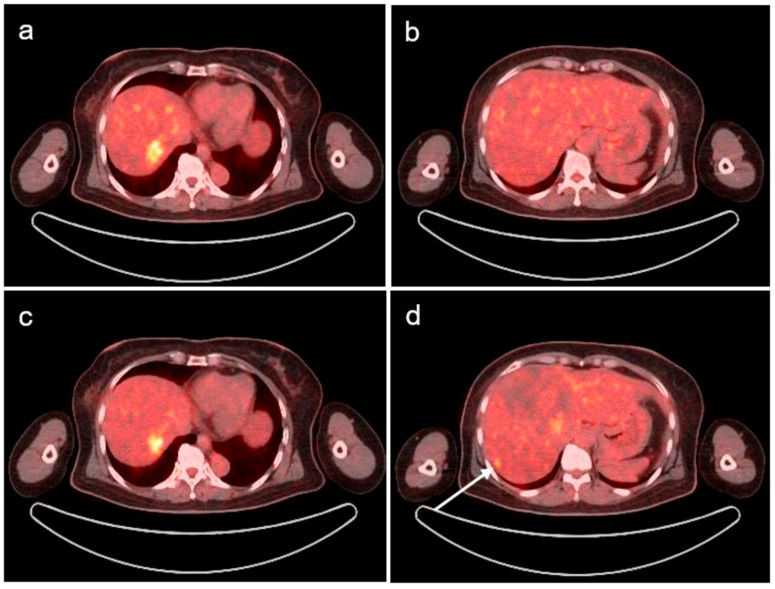
Modified dual-time-point F-18 FDG PET/CT images on routine 60 min (**a**,**b**) and early delayed 80 min (**c**,**d**) images post-F-18 FDG injection. Focal FDG uptake is seen in S6 of the liver in early delayed images compared to 60 min images after FDG injection (white arrow).

**Figure 2 diagnostics-14-01653-f002:**
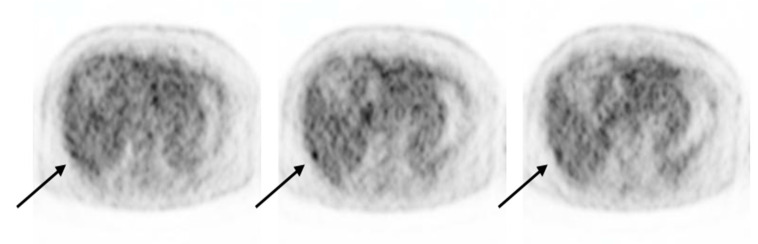
Axial PET images (top-to-bottom) show increased focal FDG uptake in segment 6 of the liver across consecutive slices (black arrow).

**Figure 3 diagnostics-14-01653-f003:**
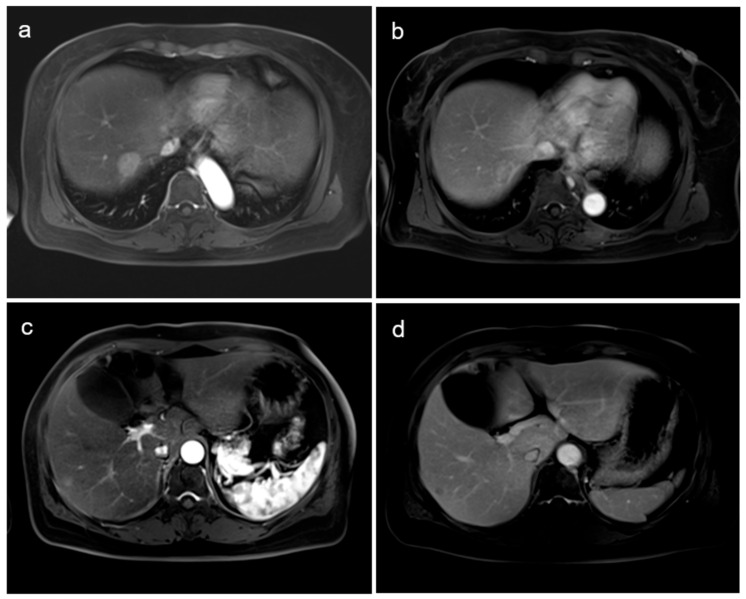
Transverse post-gadolinium T1-weighted hepatic arterial dominant phase (**a**,**c**) and hepatic venous phase (**b**,**d**) images of MRI. Two lesions of segment 7 and segment 6 showing arterial enhancement and delayed washout are seen.

## Data Availability

The data presented in this study are available on request from the corresponding author, H.-S.S.

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
