# Peer review of "Detection of Hepatic Metastasis from Early Delayed Images of Modified Dual-Time-Point F-18 FDG PET/CT Images in a Patient with Breast Cancer"

_diagnostics, 2024, doi:10.3390/diagnostics14151653_

Round 1

Reviewer 1 Report

Comments and Suggestions for Authors

Dear Authors,

I commend you for your case report, and appreciate the relevance of the message you are trying to convey. However, the images you shared are not convincing in regards to the detection of a new lesion in the early delayed phase. The image you display for evidence (Figure 1 B and D) does not show definitive evidence supporting the decision to label that focus of uptake as a lesion as I could make the case that I see a similar focus of uptake in the caudate lobe near the IVC. If you will provide an image to support your hypothesis that early delayed imaging help you detect that lesion (which could have been retrospectively identified after the MRI as MR liver would have been indicated given its definite superiority for liver MTS when compared to CT and PET/CT) the images you provide must be impeccable and without room for doubt.

Otherwise, the case report is well written, has biopsy proof (or at least it is stated) that the segment 6 lesion is a mts and is relevant as a hypothesis generation report, if the images were to show that.

Comments on the Quality of English Language

The quality of English is acceptable for a case report

Author Response

Response to Comments from Reviewer # 1

I commend you for your case report, and appreciate the relevance of the message you are trying to convey. However, the images you shared are not convincing in regards to the detection of a new lesion in the early delayed phase. The image you display for evidence (Figure 1 B and D) does not show definitive evidence supporting the decision to label that focus of uptake as a lesion as I could make the case that I see a similar focus of uptake in the caudate lobe near the IVC. If you will provide an image to support your hypothesis that early delayed imaging help you detect that lesion (which could have been retrospectively identified after the MRI as MR liver would have been indicated given its definite superiority for liver MTS when compared to CT and PET/CT) the images you provide must be impeccable and without room for doubt.

Otherwise, the case report is well written, has biopsy proof (or at least it is stated) that the segment 6 lesion is a mts and is relevant as a hypothesis generation report, if the images were to show that.

Response: Thank you for your comments.

Although the uptake of the lesion in segment 6 of the liver was mildly increased, it was observed on several slices and could be differentiated from the surrounding liver uptake.

A biopsy of the lesions in segment 6 of the liver was attempted, but they were small and were not visualized on ultrasound. Therefore, a biopsy of the lesion in segment 8 was performed.

We have mentioned these in the text and added a better image to illustrate these findings.

(Page 1, lines 35-36 and Figure 2) Although this lesion had a mild degree of FDG uptake, it was observed on several slices, unlike the surrounding heterogeneous hepatic uptake (Figure 2).

(Page 1, lines 45 – Page 2, lines 47) Since the lesion in segment 6 was too small to be observed on liver ultrasound, ultra-sound-guided liver biopsy was performed only on the lesion of segment 8.

Thank you for your insightful comments.

Reviewer 2 Report

Comments and Suggestions for Authors

In their report Lee and Song report a rare case of focal FDG uptake in the liver observed during an early delayed scanning, in a 53-year-old woman diagnosed with breast cancer.

The reported technique is extremely innovative as it allows highlighting metastatic lesions with short scanning times and avoids additional radiation exposure compared to classical methods.

This work is complete for publication.

I have a question/curiosity to ask the authors: does the reported technique have tumors for which it can be preferentially applied (or with which it can give better results)?

Author Response

Comments

I have a question/curiosity to ask the authors: does the reported technique have tumors for which it can be preferentially applied (or with which it can give better results)?

Response: Thank you for your question. In our institution, modified delayed PET/CT is used to help differentiate between tumors and inflammation, and research is underway.

Thank you.

Reviewer 3 Report

Comments and Suggestions for Authors

The authors of the manuscript titled (Manuscript ID: diagnostics-3102208) “Detection of hepatic metastasis on early delayed images of modified dual-time-point F-18 FDG PET/CT images in a patient with breast cancer” (type of manuscript: Interesting Images) present a rare case of focal FDG uptake in the liver observed during a modified dual-time-point F-18 FDG PET/CT, so-called early delayed scanning, in a 53-year-old woman diagnosed with breast cancer. The purpose of the manuscript is a current topic and presents some interesting diagnostic images.

 Major Considerations

- The images are interesting and adequately illustrate the objective of the manuscript.

 Minor considerations

-In my opinion, for readers who are not used to the technique called early delayed scanning, the authors should include an explanatory paragraph that allows them to know exactly the objective of the images presented. In that sense they could use some of the arguments described in their previous work referenced as [3].

On line 44 they could include a text similar to that previously described by the authors [3] since it would significantly help to adequately assess the scope of the images and the proposed technique:

“Over the last 10 years, there have been many reports that DTP F-18 FDG PET can be useful for discriminating between benign and malignant lesions, as well as to evaluate equivocal metastatic lesions [26]. These studies were based on the results of previous reports that FDG uptake continues to increase in malignant tumors for several hours after injection, while the uptake in benign lesions decreases or remains stable over time. However, a conventional DTP F-18 FDG PET/CT approach, which requires delayed acquisition at 120–180 min after FDG injection, involves a degree of inconvenience. Additional time is needed to obtain the delayed test (approximately 1–2 h) and additional radiation exposure occurs during the acquisition of delayed images by CT for attenuation correction. An early delayed DTP F-18 FDG PET/CT has been proposed to overcome these problems. mDTP F-18 FDG PET/CT enables lower radiation exposure to patients and shortens the scanning time compared to conventional DTP F-18 FDG PET/CT.”

Author Response

Response to Comments from Reviewer # 3

Major Considerations

- The images are interesting and adequately illustrate the objective of the manuscript.

Response: Thank you for your comment.

Minor considerations

-In my opinion, for readers who are not used to the technique called early delayed scanning, the authors should include an explanatory paragraph that allows them to know exactly the objective of the images presented. In that sense they could use some of the arguments described in their previous work referenced as [3].

On line 44 they could include a text similar to that previously described by the authors [3] since it would significantly help to adequately assess the scope of the images and the proposed technique:

Response: Thank you for your comments.

As you suggested, we have added information about modified delayed PET/CT to the text.

(Page 2, lines 49-58) Many reports have indicated that DTP F-18 FDG PET may help distinguish benign from malignant lesions and evaluate uncertain metastatic lesions, particularly hepatic metastases; these are based on previous studies showing that FDG uptake increases in malignant tumors for several hours after injection, while uptake in benign lesions de-creases or remains stable [1-4]. Modified DTP F-18 FDG PET/CT images performed 80 minutes after F-18 FDG injection, so-called early delayed images, have been recently reported in several articles [5-7]. This technique is advantageous because compared to the conventional DTP F-18 FDG PET/CT wherein there is delayed acquisition at 120–180 minutes after FDG injection, it has a shorter scanning time and avoids additional radiation exposure.

Thank you for your insightful comments.